# 100th Anniversary of Brillouin Scattering: Impact on Materials Science

**DOI:** 10.3390/ma15103518

**Published:** 2022-05-13

**Authors:** Seiji Kojima

**Affiliations:** Division of Materials Science, University of Tsukuba, Tsukuba 305-8573, Japan; kojima@ims.tsukuba.ac.jp; Tel.: +81-29-874-7728

**Keywords:** Brillouin scattering, elastic properties, acoustic phonon, central peak, crystal, glass, liquid

## Abstract

L. Brillouin predicted inelastic light scattering by thermally excited sound waves in 1922. Brillouin scattering is a non-contact and non-destructive method to measure sound velocity and attenuation. It is possible to investigate the elastic properties of gases, liquids, glasses, and crystals. Various kinds of phase transitions, i.e., liquid–glass transitions, crystallization, polymorphism, and denaturation have been studied by changing the temperature, pressure, time, and external fields such as the electric, magnetic, and stress fields. Nowadays, Brillouin scattering is extensively used to measure various elementary excitations and quasi-elastic scattering in the gigahertz range between 0.1 and 1000 GHz. A brief history, spectroscopic methods, and Brillouin scattering studies in materials science on ferroelectric materials, glasses, and proteins are reviewed.

## 1. Introduction

The well-known method to measure sound velocity is the ultrasonic method. Piezoelectricity was discovered by Jacques and Pierre Curie in 1880 [1]. Piezoelectric quartz transducers enable the detection of sound waves. However, the ultrasonic method cannot detect thermally excited sound waves due to their very weak intensity. Louis Brillouin predicted inelastic light scattering by thermally excited sound waves in 1922 [2]. It was two years before the discovery of the Bose-Einstein statistics in 1924 [3] which defined the population of thermally excited acoustic phonons. The inelastic light scattering by sound waves is called Brillouin scattering and it is a powerful tool to investigate the elastic properties of materials as a non-contact and non-destructive method.

The binding of materials involves the attractive and repulsive interactions between atoms and molecules. These interactions play the dominant roles of stability in polymorphism, amorphization, non-equilibrium states, and phase transitions. Elasticity is closely related to binding and the experimental study of elastic constants has given new insights into materials science. Various methods were developed to measure elastic constants. The time-domain and frequency-domain ultrasonic methods have been extensively used. The former ultrasonic pulse-echo method [4] measures the delay time of an ultrasonic pulse and determines the sound velocity which is related to the elastic constant and density. The latter frequency-domain resonant ultrasonic spectroscopy (RUS) measures a series of acoustic resonance frequencies of a sample [5,6]. The operating frequency of these methods is in the kilohertz to megahertz range and to measure a small sample of less than 1 mm is difficult. The mechanical contact between a sample and an ultrasonic transducer is necessary.

In contrast, Brillouin scattering is inelastic light scattering without any contact to a sample that is observed. The nondestructive measurement is possible due to the relatively weak power of the incident light. Consequently, Brillouin scattering spectroscopy is a powerful tool to study the elastic properties of various materials. Nowadays, the term Brillouin scattering is extensively used as the inelastic light scattering in the gigahertz range between 0.1 and 1000 GHz. Due to the non-contact method, phase transition, liquid–glass transition, and polymerization have been studied by changing the temperature, pressure, time, and external fields such as the electric, magnetic, and stress fields. Recently, the exciting light source was extended to UV and X-rays, which covers the terahertz-nanometer range. Brillouin scattering imaging or Brillouin microscopy enables the detection of elastic heterogeneity, which is useful for in-vivo diagnostics. The investigations of various excitations, not only acoustic phonon in the bulk and surfaces, but also magnon in magnetic materials, fracton in fractal structures, etc., give new insights into physics, chemistry, materials science, mineralogy, biology, pharmacy, medical science, and engineering.

In this review paper, a brief history of Brillouin scattering and overview of Brillouin scattering spectroscopy are described in Section 2. The various applications for materials science are discussed in Section 3, and the summary in Section 4.

## 2. Brief History and Overview of Brillouin Scattering Spectroscopy

The experimental observation of Brillouin scattering was reported eight years after the theoretical study by L. Brillouin in 1922 [2]. In 1930, E. Gross observed the inelastic light scattering from acoustic oscillations of highly scattering liquids, such as toluene and benzene at right scattering geometry using 4358 A monochromatic light from a mercury lamp and high-resolution 30 steps echelon grating [7]. The history timeline related to Brillouin scattering in the early days is listed in Table 1.

The Brillouin scattering process among the incident and scattered light and the periodic modulation of density by sound waves in a medium is shown in Figure 1. Sound waves are thermally excited and their density fluctuations cause the periodic modulation of the refractive index. An incident monochromatic light is scattered by moving periodic modulations in opposite directions.

According to the energy and momentum conservations, it holds that
(1)$$\nu_{B}=\nu_{i}-\nu_{s},\;{\vec{k}}_{i}={\vec{k}}_{s}\pm \vec{q}$$
where ${\vec{k}}_{i}$, ${\vec{k}}_{s}$, and $\vec{q}$, $\nu_{B}$ are the wavevectors of the incident photon, scattered photon, and acoustic phonon, respectively. *ν*_i_, *ν*_s_, and *ν*_B_ are the frequency of incident light, scattered light, and the acoustic phonon, respectively.

The Brillouin scattering spectrum of the liquid phase of propylene glycol observed by a Fabry-Perot interferometer (FPI) [8,9] is shown in Figure 2. A central intense peak corresponds to Rayleigh scattering and the doublet peaks correspond to Brillouin scattering by sound waves.

The differential cross section of the Brillouin scattering of a fluid is given by
(2)$$\frac{d\sigma }{d\Omega }\propto \beta_{S}\rho^{2}{\left(\frac{\partial \chi }{\partial \rho }\right)}_{S}^{2}+\frac{T}{\rho C_{p}}{\left(\frac{\partial \chi }{\partial \rho }\right)}_{p}^{2},$$
where *β*_S_ is the adiabatic compressibility, ρ is the density,χ is the susceptibility, and *C*_p_ is the heat capacity at a constant pressure. *S*, *T*, *p* are the entropy, temperature, and pressure, respectively [13]. The first term on the right side corresponds to the propagating density fluctuations of the sound waves at a constant entropy and it is related to a doublet of Brillouin shifts by sound waves. The second term corresponds to non-propagating entropy fluctuations at a constant pressure, and it is related to a Rayleigh peak. In a fluid, only the longitudinal acoustic (LA) mode is observed, while in a solid, the transverse acoustic (TA) mode is also observed due to the existence of a shear modulus. Brillouin scattering of KCl, RbCl, and KI crystals with a cubic system was studied [14]. The scattering intensity of a cubic crystal by the LA and TA modes, which propagate along the [100] axis is given by
(3)$$I_{VV}\propto \frac{{\left(p_{12}\right)}^{2}}{c_{11}},\;I_{VH}\propto \frac{{\left(p_{44}\right)}^{2}}{2c_{44}},$$
respectively. VV denotes that the polarization of both the incident and scattered light is vertical and VH denotes that the polarization of the incident light is vertical while that of the scattered light is horizontal. The *p*_ij_ and *c*_ij_ are the elasto-optic and elastic stiffness constants, respectively. The displacements of the LA and TA modes are [100] and [001] directions, respectively. The method for the determination of the elastic and photoelastic constants was analyzed for all the crystal systems, except for the low-symmetry systems [15].

In the visible light excitation, the frequency of the scattered acoustic phonon (Brillouin shift) is usually from 10 to 200 GHz. To measure the Brillouin scattering spectra, a spectrometer with a high resolution of 0.1 GHz or 0.01 A is necessary. Therefore, it is difficult to observe by using a conventional grating monochromator for Raman scattering spectroscopy. For the first observation of Brillouin scattering, the higher order diffraction of an echelon grating was used [7], while in most experiments, a Fabry-Perot interferometer (FPI) [8,9] was used. The FPI consists of two parallel partially reflecting surfaces with the distance *d* to observe the interference by the multi-reflection between reflecting surfaces. The free spectral range (FSR) of FPI is 1/2*d*, for example, FSR = 30 GHz for *d* = 5 mm. Figure 2 shows the FPI spectrum of propylene glycol with FSR = 30 GHz [16]. On the horizontal axis, the wavelength of the scattered light is converted to the frequency shift from the frequency of the incident monochromatic light. The zero-frequency shift is adjusted to a Rayleigh peak and the cyclic Brillouin scattering spectra appears with the period of 30 GHz. This spectrum was measured by an angular dispersive FPI (ADFPI), in which the finess = 100 is possible using a solid etalon [17,18]. In ADFPI, the angular dispersion of a solid etalon was used, and the one-dimensional output signals were detected by a highly sensitive CCD detector.

During the early stage, the output wavelength of the FPI was scanned by the change of in the refractive index induced by the release of air into a sealed FPI chamber in a vacuum. The scattered light was analyzed using a pressure-scanned Fabry-Perot interferometer (FPI) and “photon-counting” electronics. For crystals with intense elastic scattering, a tandem FPI was used to measure an improper ferroelectric β-Gd_2_(MoO_4_)_3_ crystal [19] and a double-pass FPI was used to measure a proper ferroelastic LiNH_4_C_4_H_4_O_6_-H_2_O crystal [20]. After scanning an etalon spacing by piezoelectric elements was developed, the long acquisition time of weak signals using a multi-pass FPI became possible. For the broadband spectra with a high contrast, the tandem multi-pass FPI was developed to measure a wide frequency range up to 1000 GHz using two FPIs with slightly different FSRs [21]. By the connection to a Raman scattering spectrum measured by a grating-spectrometer, a very broadband spectrum is available between 1 GHz and 120 THz including both acoustic and optical phonons.

A typical broadband inelastic light scattering spectrum of a ferroelectric Pb(Sc_1/2_Ta_1/2_)O_3_ crystal with the perovskite structure between 1 GHz (0.033 cm^−1^) and 36 THz (1200 cm^−1^) is shown in Figure 3 [22]. It consists of a Brillouin scattering spectrum measured by a 3 + 3 pass tandem Fabry-Perot interferometer and a Raman scattering spectrum measured by a triple grating spectrometer. In the low-frequency range below 300 GHz (10 cm^−1^), the peak of an acoustic phonon is observed, while in the higher frequency above 10 cm^−1^, several peaks of optical phonons are observed. In addition, a broad Rayleigh wing of quasi-elastic scattering by the relaxation mode related to polarization fluctuations appears below the lowest-frequency optical phonon. It is called a central peak (CP) and it plays an important role in the dynamical properties in order-disorder phase transitions and liquid–glass transitions.

The double monochromator double pass (DMDP) spectrometer allows the simultaneous measurement of Brillouin scattering and Raman scattering by using a high-resolution echelle grating. Since the radiation effectively travels a distance of 32 m, a DMDP spectrometer is large and delicate [23].

Usually, the measurements of FPI with a single channel detector, such as photomultiplier, or photodiode, require a long acquisition time. For a rapid measurement, the angular dispersive FPI (ADFPI) was developed by using a solid etalon and a highly sensitive multichannel detector for the rapid spectrum acquisition. The acquisition time of 1 s or less is possible, which is much shorter than that of a scanning FPI [17]. Using a solid etalon with a high reflectivity (99.5%), the high resolution and high contrast spectrum was achieved. The finesse, the inverse of the resolution, reached 100 [18]. A virtually imaged phased array (VIPA) is another type of angular dispersive FPI [24]. VIPA coupled to a CCD detector enables Brillouin scattering imaging, and the in situ biomechanical measurement of the crystalline lens in a mouse eye using Brillouin shifts. [25]. The high performance of Brillouin microscopy was reported using microwave-induced acoustic fields in a piezoelectric LiNbO_3_ [26]. The possibility was shown to detect the microwave-induced bulk as well as surface acoustic waves.

As an ultrasonic method, scanning acoustic microscopy (SAM) was developed using a finely focused ultrasonic beam in the GHz range [27]. SAM is an imaging system with a resolution comparable to an optical microscope. Another advantage of the SAM is the determination of the phase velocity of surface acoustic waves by measurement of the periodicity of a *V(z)* curve, where *V* is the output signal of an acoustic lens and *z* is the distance between a sample surface and a lens [28]. By using the ultrasonic frequency of 8 GHz and liquid helium as the coupling medium at 0.2 K, the resolution of 200 A was reported [29]. The improvement of the special resolution better than an optical microscopy may be expected for Brillouin imaging.

The stimulated Brillouin scattering (SBS) was developed by Chiao, Townes, and Stoicheff [30]. SBS is a stimulated process of light scattering that occurs when the intensity of the light field itself affects the propagating medium. The rate of Brillouin scattering is markedly enhanced by the powerful coherent laser light. SBS has recently become relevant in the optical fiber industry [31,32]. The time-domain Brillouin scattering (TDBS) has been used for the detection of inhomogeneity [33].

Nowadays, Brillouin scattering is extensively used in the studies of various kinds of materials such as gases [31], colloids [34], multilayers [35], ceramics [36], amino acids [37], biological tissues [38,39], insects [40], and fossils [41]. Brillouin scattering has been also used as inelastic light scattering in the gigahertz range for the investigations of various elementary excitations, not only for the acoustic phonon, but also the magnon [42,43], fracton [44], roton [45], ripplon [46], and soliton [31] have been studied. This knowledge offers new insights into physics, chemistry, materials science, mineralogy, biology, pharmacy, medical science, and engineering.

## 3. Brillouin Scattering Spectroscopy in Materials Science

The combination of a tandem multi-pass FPI with an optical microscope enables the measurements of a very small material or a very small area of a sample as shown in Figure 4 [47]. The monochromatic light beam from a single frequency CW laser is focused into a specimen to be observed through the objective lens of a reflection optical microscope. The scattered light from a specimen is collected by the objective lens at backward scattering geometry. The scattered light is collimated and focused on the input pinhole of a Sandercock-type 3 + 3 pass tandem multi-pass FPI and detected by a photon-counting system. A compact IR furnace is connected to measure a sample at high temperatures [48]. For the measurements at low temperatures, a heating/cooling stage [49] or cryostat [50] is used. For the measurements at high pressures, diamond anvils are very convenient and powerful. The size of a sample is 0.1 mm or less as it is possible to observe Brillouin scattering using a finely focused laser beam.

For the measurements under high pressure, diamond anvils are convenient as shown in Figure 5a [51]. By shortening the distance between a pair of diamond anvils, a high pressure is generated in the sample chamber consisting of the sample and pressure transmitting medium. Figure 5b shows the pressure dependences of compressibility and Poisson’s ratio of Baltic amber determined by the LA and TA shifts up to 12 GPa [52]. Baltic amber is a natural organic glass with the high fragility index of *m* = 90. As the pressure becomes very high, the sample size becomes very small and the ultrasonic methods become difficult to measure its elastic properties.

Brillouin scattering spectroscopy is a powerful tool to investigate elastic properties and dynamical properties in the gigahertz range, and various materials have been studied. It is possible to measure a sample under extreme conditions such as high temperature up to 2000 °C [53] and high pressure up to 100 GPa [54]. In this section, a few topics regarding the Brillouin scattering studies of functional ferroelectric materials, glass-forming materials, and proteins are reviewed.

### 3.1. Ferroelectric Materials

Ferroelectricity is defined by the existence of spontaneous polarization and its switching by external electric fields. Ferroelectricity was first found in Rochel salt in 1921 by Balasec [55]. Ferroelectric crystals have no center of symmetry and their physical properties of odd order tensors, such as pyroelectric, piezoelectric [1], electro-optic, nonlinear-optic properties, and polaritons, are technologically very important [56]. They have been extensively used as functional materials. A ferroelectric material undergoes a paraelectric-ferroelectric phase transition at its Curie temperature upon heating. The highest Curie temperature is 1342 °C for Sr_2_Nb_2_O_7_ with a perovskite-slab structure [57]. The high tunability of the Curie temperature from such a high temperature down to −107 °C is possible by the control of the composition of Sr_2_(Nb_1-x_Ta_x_)_2_O_7_. It is also technologically important [58].

Most ferroelectric materials are oxygen octahedra ferroelectrics such as perovskite, layered perovskite, perovskite slab, and tungsten bronze (TB) structures. In this section, normal and relaxor ferroelectrics with the TB structure are described. The projection on the (001)-plane of the oxygen octahedral framework of the TB structure is shown in Figure 6 [59]. There are three types of A1, A2, and C sites. B1 and B2 are the central sites of the oxygen octahedra. A tetragonal unit cell includes two A1 sites, four A2 sites, four C sites, two B1 sites, eight B2 sites and thirty oxygens. The site occupancy formula of the tetragonal TB structure is (A1)_2_(A2)_4_(C)_4_(B1)_2_(B2)_8_O_30_. The smallest triangular C sites are occupied only by Li ions. The ferroelectrics with filled TB, such as Ba_2_NaNb_5_O_15_ (BNN), undergo a normal ferroelectric phase transition of which dielectric constant along the *c*-axis obeys the Curie-Weiss law. However, the ferroelectrics with an unfilled TB structure, such as Sr_1-x_Ba_x_Nb_2_O_6_ (SBN100x), show relaxor behaviors with a broad dielectric anomaly and remarkable frequency dispersion [60].

BNN is important in the field of telecommunications due to its superior electro-optical and nonlinear optical properties such as second harmonic generation, and its resistance to optical damage is high. The A1 and A2 sites are fully filled by Na and Ba, respectively, and there is no vacancy. The B1 and B2 sites are fully occupied by Nb, while the C sites are vacant. BNN is called a filled TB structure with no charge disorder at A1 and A2 sites. BNN undergoes successive phase transitions at 575 and 290 °C. The higher one is associated with the ferroelectric Curie temperature at 575 °C and its crystal symmetry changes from prototypic *4/mmm* to ferroelectric tetragonal *4 mm* with a spontaneous polarization *P*_3_ along the *c*-axis. Upon the symmetry changes from tetragonal to incommensurate (IC) orthorhombic *mmm* systems at 290 °C, the *a* and *b* axes are rotated 45° along the *c*-axis as shown in Figure 6. The modulation direction of the IC wave vectors is along the *a* and *b* axes of the orthorhombic coordinate [59].

#### 3.1.1. Elastic Anomaly

The sound velocity *V* is determined by the frequency shift *ν*_B_ in a Brillouin scattering spectrum using the following equation:(4)$$V=\frac{\lambda_{i}v_{B}}{2\;n\sin\frac{\theta }{2}}$$
where, *λ_i_*, *θ*, and *n* are the wavelength of the incident beam, the scattering angle, and the refractive index of the sample, respectively. The TA and LA velocities are determined from the TA and LA frequency shifts, respectively.

The attenuation α is determined by,
(5)$$\alpha =\frac{\pi \Gamma }{V}$$
where Γ is FWHM of the TA or LA peak.

The temperature dependence of the LA peaks observed at backward scattering using a *c*-plate is shown in Figure 7a [61]. The wave vector of the observed LA mode is parallel to the c-axis. The LA peaks shows softening from 800 °C to *T*_C_ = 575 °C. The temperature dependences of the LA velocity and attenuation determined from the LA shift and its FWHM using Equations (1) and (2), respectively, are shown in Figure 7b.

Since the polarization fluctuations along the *c*-axis couple with the strain along the *c*-axis by the electrostrictive coupling, remarkable changes were observed in the temperature dependences of the sound velocity and attenuation of the LA mode, which propagate along the *c*-axis. Such a sharp elastic anomaly in the vicinity of the Curie temperature is the typical nature of a normal ferroelectric phase transition. In contrast, the diffusive changes were observed in a relaxor ferroelectric phase transition with random fields [22,62]. Using the power law behaviors of the sound velocity and attenuation above the Curie temperature, the dimensionality of fluctuations of the order parameters and the order-disorder nature were discussed [63,64].

#### 3.1.2. Critical Slowing Down

Ferroelectrics are divided into two types, namely, the displacive type and the order-disorder type according to the difference in the mechanism upon the appearance of a spontaneous polarization. In the displacive type, an infrared active soft-optic mode exists and a spontaneous polarization appears by the freezing of a soft-mode displacement. However, in the order-disorder type, the relaxation time of the polarization fluctuations along a ferroelectric axis diverges at the Curie temperature, and the critical slowing down towards the Curie temperature is observed. No soft optic mode was observed in the ferroelectrics with the TB structure, however, an intense central peak (CP) of BNN was observed in the vicinity of the Curie temperature in the broadband Brillouin scattering spectra as shown in Figure 8a [61].

The temperature dependence of the relaxation time determined by the width of a broad central peak observed at backward scattering in the *a*-plate of BNN is shown in Figure 8b [61]. The discontinuity of the relaxation time at the Curie temperature indicates the first order phase transition. The critical slowing down towards the Curie temperature was clearly observed. The divergence of the relaxation time of the polarization fluctuations along the ferroelectric *c*-axis is evidence for the order-disorder nature of the ferroelectric phase transition of BNN. A similar critical slowing down was also observed in ferroelectric LiTaO_3_ [65] and BaTi_2_O_5_ [66]. However, in LiTaO_3_, the soft optic mode was also observed far below the Curie temperature. Therefore, the crossover from the displacive type to order-disorder one occurs, and it is related to the single minimum potential of Nb ions and double minimum potential of Li ions along the ferroelectric *c*-axis. In contrast, in Sr_2_Nb_2_O_7_, the soft optic mode was clearly observed at all the temperatures, and it is evidence of the pure displacive type [58].

### 3.2. Relaxor Ferroelectrics

In normal ferroelectrics, the dielectric constant along a ferroelectric axis shows a sharp peak at the Curie temperature with no frequency dispersion. However, the dielectric constant of relaxor ferroelectrics shows a broad diffusive peak with a remarkable frequency dispersion [67]. The origin of the relaxor natures is the structural disorders, such as the random occupancy of heterovalent cations, vacancies, and frustration of competing interactions. Typical relaxor ferroelectrics are Pb(B_1/3_B’_2/3_)O_3_ [62], Pb(B_1/2_B’_1/2_)O_3_ [22], with the perovskite structure and M_1-x_Ba_x_Nb_2_O_6_ (M = Ca, Sr) with the unfilled TB structure [64]. Relaxor ferroelectrics (REFs) have attracted much attention due to their colossal piezoelectric effect and their wide variety of applications in piezoelectric devices. Pb-based multidimensional REFs with the perovskite structure have been extensively studied [22,62], whereas the understanding of uniaxial relaxors is still unresolved.

#### 3.2.1. Characteristic Temperatures of Relaxor Ferroelectrics

Regarding the physical properties of the REFs, the polar nanoregions (PNRs) with a polar structure play an important role. Figure 9 shows the sequence of the characteristic temperatures of a relaxor ferroelectric phase transition, *T*_B_⇒ *T** ⇒ *T*_C_ ⇒ *T*_f_ [22,62]. When a sample is cooled from a very high temperature, at the Burns temperature, *T*_B_, dynamic PNRs nucleate in a paraelectric phase with a center of symmetry. The *T*_B_ is typically a few hundred degrees above the ferroelectric Curie temperature, *T*_C_. With further cooling from *T*_B_, a dynamic central peak in an inelastic scattering spectrum is observed by thermally fluctuated flipping of the local polarizations in the PNRs. An elastic anomaly is observed in the vicinity of *T*_C_ by the local piezoelectric coupling in the PNRs. In the temperature range between *T*_B_ and *T*_C_, a dynamic-to-static transition of the PNRs occurs at the intermediate temperature *T**. All the PNRs are completely frozen at the freezing temperature, *T*_f_, which is the Vogel-Fulcher temperature [22]. At *T*_C_, the polarization fluctuations of the PNRs are frozen into nanodomain structures, because the growth into macrodomains is blocked by the random fields (RFs). Consequently, the nanodomain structures are nonequilibrium states, and gradually change into equilibrium macrodomain structures by aging with a long relaxation time. For the investigation of the REFs, Brillouin scattering spectroscopy is a powerful tool to investigate the elastic anomaly, relaxation processes, and aging versus their temperature, pressure, time, and electric field dependences [22,62].

In the unfilled TB structure, there are vacancies in the A1 and A2 sites, which cause the charge disorder. This is due to the RFs of the relaxor nature. Sr*_x_*Ba_1−*x*_Nb_2_O_6_ (SBN100*x*), is one of the technologically important uniaxial REFs due to its very high electro-optic and pyroelectric coefficients, making it useful for applications such as sensors and data storage. In the SBN, the largest A2 sites are occupied by the divalent Ba and Sr ions, while the A1 sites are occupied only by the smaller Sr ions. One-sixth of all the A sites (A1 + A2 sites) are vacancies, which are the main origin of the RFs. The relaxor nature of SBN was described by the random field Ising Model [68]. With the increasing Sr/Ba ratio, SBN shows a crossover from normal to relaxor ferroelectrics by the enhancement of the charge disorder. Ca*_x_*Ba_1-*x*_Nb_2_O_6_ (CBN100*x*) has also attracted attention by its similar relaxor nature and high Curie temperatures, which are important for application in a high-temperature environment [69]. The crossover from normal to relaxor ferroelectrics has been studied in SBN100x and CBN100*x*. In SBN26 with weak RFs, a sharp elastic anomaly was observed in the vicinity of the Curie temperature similar to the anomaly of BNN. However, in SBN80 with strong RFs, a broad and diffuse elastic anomaly was observed [63].

For the optical applications of the SBN, in most cases, the congruent melt SBN61 has been used, because large and high-quality single crystals are available by the Czochralski method. The relaxor ferroelectric SBN61 with a strong charge disorder undergoes a diffuse phase transition upon cooling from a prototypic tetragonal *4/mmm* phase to a ferroelectric tetragonal *4mm* phase with a spontaneous polarization along the *c*-axis [60]. The acoustic phonons are sensitive to the three characteristic temperatures due to the scattering caused by the PNRs. The temperature dependence of the LA attenuation of SBN61 is related to the characteristic temperatures as shown in Figure 10 [70]. The increase in the LA width was observed near *T*_B_. Below *T**, the scattering of the LA mode by the static PNRs causes a significant increase in the LA width. At the *T*_C_, the increase in the scattering by the PNRs was stopped by the freezing of the PNRs into ferroelectric nanodomain structures [71]. The three characteristic temperatures, *T*_B_ = 350 °C, *T** = 190 °C, and *T*_C_ = 72 °C, were determined. *T*_B_ and *T** are very close to those of SBN75, therefore, *T*_B_ and *T** may be common to all the compositions of the relaxor SBN100*x*.

#### 3.2.2. Electric Field Effects on Relaxor Ferroelectrics

The electric field dependence of the LA velocity along the *c*-axis of SBN61 after zero field cooling (ZFC) is shown in Figure 11 [70]. The electric field along the ferroelectric *c*-axis was first applied to a ZFC crystal in the nonequilibrium nanodomain state, “Nano”. The initial LA velocity of 5270 m/s of “Nano” gradually increased up to 5430 m/s at Es = 1.7 kV/cm, then the splitting of a LA peak occurred up to Es = 2.4 kV/cm. Another high frequency peak produced a higher velocity of 5760 m/s in the equilibrium macrodomain state, “Macro”, which is a nearly single domain state and only macrodomains exist. In the field range between Es = 1.7 kV/cm and Ef = 2.4 kV/cm, the “Nano” and “Macro” states coexist, and it is called the “Mix” state. For a further increase in the electric field, the splitting of the LA peaks disappeared, and only a single higher frequency peak appeared. Above Ef, the high LA velocity of 5760 m/s showed only a “Macro” state up to 3.6 kV/cm. When the electric field decreased from E = 3.6 kV/cm, the velocity kept the high value and a “Macro” state remained down to E = 0 kV/cm.

In order to clarify the temperature dependence of the three different states, “Nano”, “Mix”, and “Macro” in Figure 11, heating and cooling with and without an electric field were investigated. The LA velocity was first measured during field heating (FH) from the “Macro” state at room temperature up to *T** as shown in Figure 12. A large thermal hysteresis was then observed below *T*_C_ during the ZFC. A similar hysteresis was also reported for the Pb-based REFs [72]. At room temperature, it was found that the LA velocity after the ZFC is much lower than after FH, and the “Nano” state appears. Next, after annealing for a month at room temperature, the velocity increased by about 2%, and the “Mix” state appeared as the intermediate state with a mixture of nano- and macrodomain states. The slow aging from the “Nano” to “Macro” states was also investigated at room temperature. For the complete recovery to the “Macro” macrodomain state at room temperature, it is necessary to keep a ZFC crystal for more than 10 years. The temperature, electric field dependences, and aging were also reported for SBN40 and SBN70 [73,74].

### 3.3. Liquid, Glass, and Crystalline States

#### 3.3.1. Liquid–Glass Transition

When a liquid is cooled from a high temperature, a simple liquid is solidified into a crystalline state at its melting temperature, *T*_m_. However, a complex liquid changes into a supercooled liquid state, and with further cooling, it undergoes a liquid–glass transition into a glassy state at the glass transition temperature *T*_g_. Figure 13 shows the temperature dependence of enthalpy for a liquid (AB), supercooled liquid (BD), glassy (DE), and crystalline (CG) states, where *T*_g_, and *T*_K_ are the glass transition and Kauzmann temperatures, respectively. The enthalpy of a liquid crosses to that of a crystal at the point, F. The temperature at F is the Kauzmann temperature, *T*_K_, which is a static ideal glass transition temperature and close to the Vogel–Fulcher temperature, the dynamic ideal glass transition temperature, *T*_VF_.

When a liquid is cooled to a liquid–glass transition, various types of dynamical properties, such as a main α-relaxation, slow and fast β-processes and a boson peak were observed in the high frequency range from the mHz to THz ranges [47]. The relaxation time of the main α-structural relaxation process in a fragile liquid, such as ortho-terphenyl (OTP), diverges at *T*_g_, and obeys the Vogel–Tammann–Fulcher law reflecting the cooperative motion in a cage. However, in a strong liquid, such as a silica glass, the relaxation time obeys the Arrhenius law and no remarkable change is observed at *T*_g_.

Glycerol undergoes a liquid–glass transition at about *T*_g_ = 187 K. It is one of the typical glass-forming materials and well-known cryoprotectants due to its strong glass-forming tendency [75]. According to strong-fragile classification, it is the intermediate liquid with the fragility index, *m* = 53 [76]. Figure 14 shows several relaxation processes and vibrational dynamics in glycerol. A low-frequency Raman scattering spectrum covers a boson peak [77]. A broadband Brillouin scattering spectrum covers the high-frequency part of the α-relaxation, the γ-relaxation, the fast β-relaxation in a cage, and the low-frequency part of a boson peak [78]. The relaxation times of the α- and slow β-processes were observed by dielectric spectroscopy, which detects polar motions [79]. Heat capacity spectroscopy detects polar and nonpolar motions [80]. For understanding of the dynamical features of a liquid–glass transition, it is very important to draw a complete picture of an ultra-broadband relaxation map, which covers from very slow to very fast processes, and a boson peak.

The polarized and depolarized broadband spectra between 1 GHz and 10 THz measured by a tandem multi-pass FPI and a Raman spectrometer were studied in silica, OTP and ethanol [81]. It was found that the amplitude of this excess quasi-elastic scattering in polarized scattering is proportional to the ratio of the integral intensities of the Brillouin line and the boson peak, while the relaxation mechanisms are the same for both the polarized and depolarized spectra.

For comparison with the dynamic susceptibility predicted by the idealized mode coupling theory of a liquid–glass transition, the composite depolarized spectra were extensively studied in various glass-forming liquids [82]. In this study, the problem of a “knee” in the fragile Ca_0.4_K_0.6_(NO_3_)_1.4_ (CKN) was reported as an experimental artifact, which can be eliminated by the addition of a narrow-band dielectric filter to the Fabry-Perot optics [83]. However, for *T* that was above but close to the critical temperature of the mode coupling theory (MCT), *T*_C_, the susceptibility spectra were found to obey the MCT predictions for all the materials such as salol, propylene carbonate, glycerol, iso-propyl benzene, OTP, and CKN.

For the liquid–glass transition of symmetric molecules, a rapid cooling is necessary to avoid crystallization in a supercooled liquid state. The liquid–glass transitions of trimethylene glycol, deuterated ethylene glycol-d_4_, and poly(propylene glycol) diglycidyl ether were studied by a rapid cooling using the short acquisition time of ADFPI [17,84,85].

When a liquid is cooled at a constant pressure, the free volume decreases and a liquid–glass transition occurs. When the hydrostatic pressure is applied to a liquid at a constant temperature, the free volume decreases and a pressure-induced liquid–glass transition occurs. The pressure induced liquid–glass transitions of a few lower alcohols were studied using a DAC at room temperature and the change in the elastic properties was observed at the glass transition pressure [86,87].

#### 3.3.2. Drug Materials

The polymorphic nature of pharmaceutical materials is attributed to their good glass-forming tendency. Since nonequilibrium glassy/amorphous states have a solubility and dissolution rate that is higher than the equilibrium crystalline states, the presence of the glassy/amorphous states have the advantage in pharmaceutical materials. The liquid, glassy, and crystalline states of various drug materials, such as aspirin [88,89], ibuprofen [90,91], ketoprofen [92], and indomethacin [34,93] were studied by Brillouin scattering.

Indomethacin (IMC) [1-(p-chlorobenzoyl)-5-methoxy-2-methy lindole-3-acetic acid] is one of the non-steroidal anti-inflammatory drugs and is used for the treatment of fever, pain, and swelling [94]. The liquid, supercooled liquid, and crystalline states of IMC were studied by Brillouin scattering at the backward scattering and reflection induced ΘA scattering geometries [95]. The temperature dependences of the Brillouin scattering spectra are shown in Figure 15. The temperature dependences of the Brillouin shifts of the LA and TA modes of the liquid, glass, and γ-form crystalline IMC are shown in Figure 16. Upon heating of the glassy IMC, the elastic anomaly at the liquid–glass transition temperature *T*_g_ = 315 K was clearly observed. With further heating of the supercooled liquid state, crystallization at *T* = 410 K and melting at *T*_m_ = 443 K into an equilibrium liquid phase were also observed.

The TA mode of a glassy state was measured using reflection-induced ΘA scattering [95] as shown in Figure 17a. The observed spectrum is shown in Figure 17b. From the LA and TA Brillouin shifts, the bulk modulus *K* = 5.41 GPa and shear modulus *G* = 1.33 GPa were determined.

The temperature dependence of the LA and TA mode frequency of a γ-form IMC crystal was measured at the backward scattering geometry. The elastic constants at 296 K are shown in Table 2 [95].

### 3.4. Polymorphism, Dehydration, and Denaturation of Proteins

Brillouin scattering is a convenient method to investigate the elastic properties of protein crystals and their phase transitions or transformations. Regarding the study of tetragonal lysozyme crystals grown by the hanging-drop vapor-diffusion method, an elastic anomaly was observed at 34 °C [96]. This fact indicated a structural phase transition. The elastic properties of a tetragonal hen egg-white lysozyme (HEWL) crystal were studied in a cryoprotective solution over a wide temperature range [97]. The dehydration process of tetragonal and monoclinic lysozyme crystals grown by the two-liquids interface method [98] was studied by exposure of the crystals to open air using Brillouin scattering [99]. The LA velocity markedly increased due to the increase in intermolecular interaction between the lysozyme molecules, while the LA attenuation monotonically decreased due to the decrease in the friction generated by the mobile water. The time dependence of the LA velocity of the tetragonal HEWL at a constant temperature was measured. At 40 °C, the LA velocity markedly increased after 20 min and approached the horizontal asymptote. In contrast, the LA velocity at 30 °C moderately increased and reached a constant value that is slightly lower than that at 40 °C.

The time dependence of the LA velocity was analyzed by the Avrami–Erofe’ev model [100]. The normalized dehydration fraction is defined by
(6)$$r\left(t\right)=\frac{V\left(t\right)-V\left(0\right)}{V\left(\infty \right)-V\left(o\right)}$$

For the Avrami–Erofe’ev type reaction, it holds that
(7)$$-\ln\left\{1-r\left(t\right)\right\}=kt^{m},$$
where *k* is the dehydration rate constant, *t* is the time, and *m* is the dimension number of the reaction. Figure 18 shows the timed dependence of *r*(*t*) during dehydration at 30 °C and 40 °C. The dimension of this process is different between a slow two-dimensional process (*m* = 2) at 30 °C and a fast three-dimensional process (*m* = 3) at 40 °C in accordance with the AE model. However, for a monoclinic HEWL crystal, the *r*(*t*) was three-dimensional at 32 °C. These findings may indicate that the protein–water interaction is sensitive to the dehydration rate and the rate depends on the crystal structure.

The lysozyme and denaturing guanidine-hydrochloride solutions were studied by Brillouin scattering [95,101]. The Brillouin spectrum of the lysozyme unfolded in 6M guanidine hydrochloride contained narrow and broad quasi-elastic scattering peaks. The narrow peak corresponds to interactions of the denaturing agent with the polypeptide chain of the lysozyme. However, the broad peak is attributed to the interaction among the guanidinium (Gdm^+^) [CH_6_N_3_]^+^ cations.

## 4. Conclusions

Brillouin scattering is the inelastic light scattering in the gigahertz range from 0.1 to 1000 GHz with a high resolution. It allows the measurement of not only the elastic properties but also various elementary excitations and relaxation processes. Since its measurement is noncontact and nondestructive, it is possible to investigate the dependences of various dynamical properties on the temperature, pressure, time, and external fields. The sharp elastic anomaly and critical slowing down of a normal ferroelectric phase transition are shown. The electric field effect and aging of the non-equilibrium nanodomain states into equilibrium macrodomain states are discussed in relaxor ferroelectrics. The liquid, glass, and crystalline states of drug materials are clarified by their temperature dependence on the elastic properties. The polymorphism and denaturation of proteins are discussed by the variation in the elastic properties and quasi-elastic scattering.

## Figures and Tables

**Figure 1 materials-15-03518-f001:**
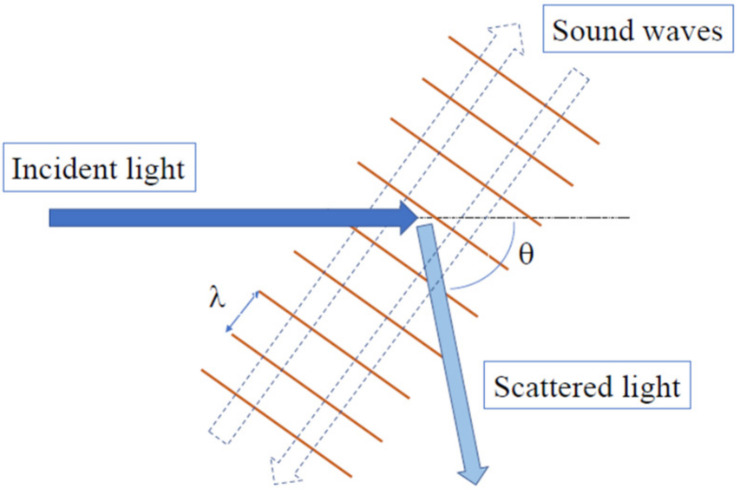
Brillouin scattering process among the incident and scattered light and periodically modulation of the density by thermally excited sound waves, where λ and θ are the wavelength of sound waves and scattering angle, respectively.

**Figure 2 materials-15-03518-f002:**
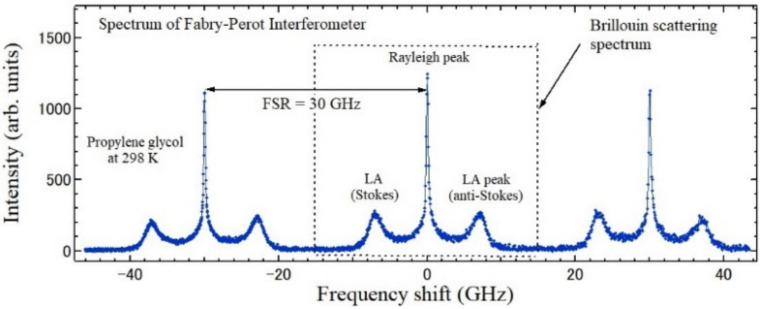
The spectrum of an angular dispersive Fabry-Perot interferometer of propylene glycol in a liquid phase at 298 K with the free spectral range of 30 GHz.

**Figure 3 materials-15-03518-f003:**
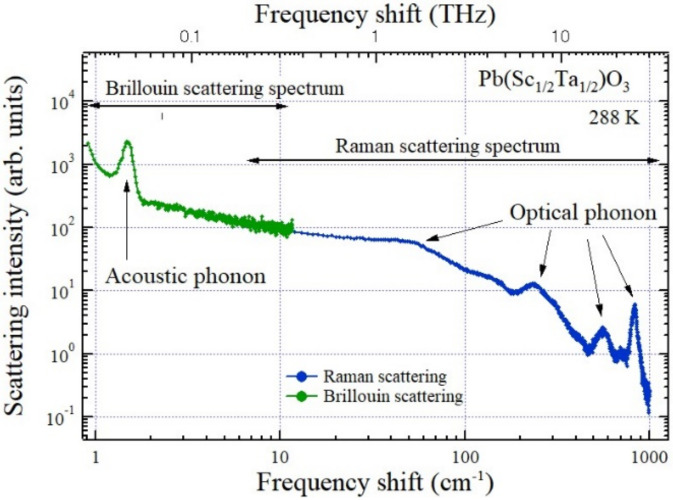
Broadband inelastic light scattering spectrum of a ferroelectric Pb(Sc_1/2_Ta_1/2_)O_3_ crystal. Brillouin and Raman scattering spectra were measured by a tandem multi-pass FPI and triple-grating spectrometer, respectively [22].

**Figure 4 materials-15-03518-f004:**
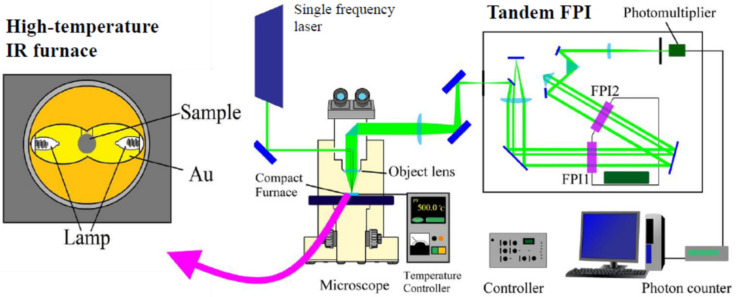
Experimental setup of micro-Brillouin scattering using a tandem multi-pass Fabry-Perot interferometer [48].

**Figure 5 materials-15-03518-f005:**
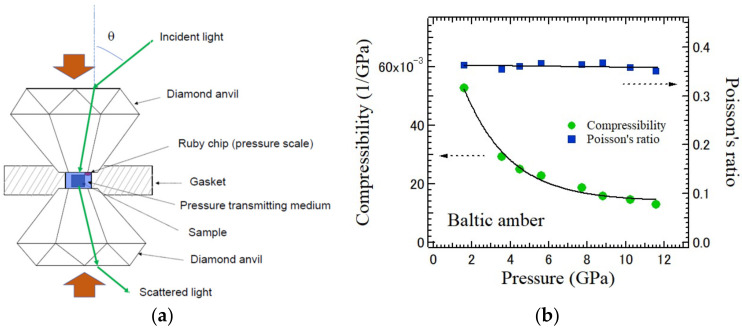
(**a**) Brillouin scattering experiment at high pressures using a diamond anvil cell. (**b**) Pressure dependences of compressibility and Poisson’s ratio of the Baltic amber measured by Brillouin scattering [51].

**Figure 6 materials-15-03518-f006:**
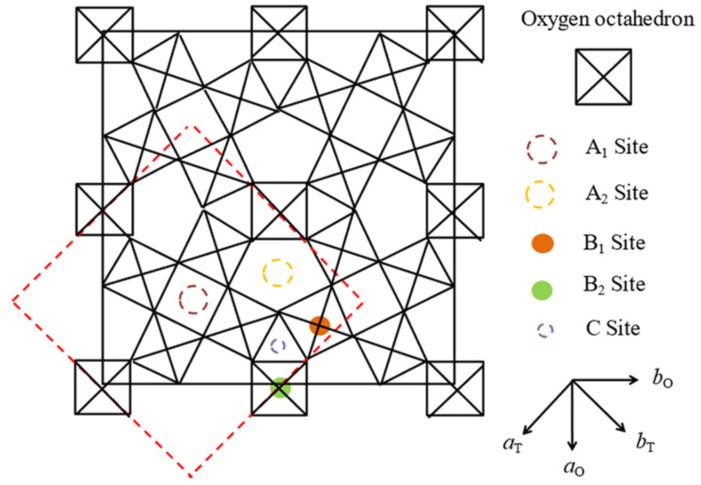
Projection of tungsten bronze structure on the *c*-plane. Tetragonal and orthorhombic unit cells are shown by dotted and solid lines, respectively [60].

**Figure 7 materials-15-03518-f007:**
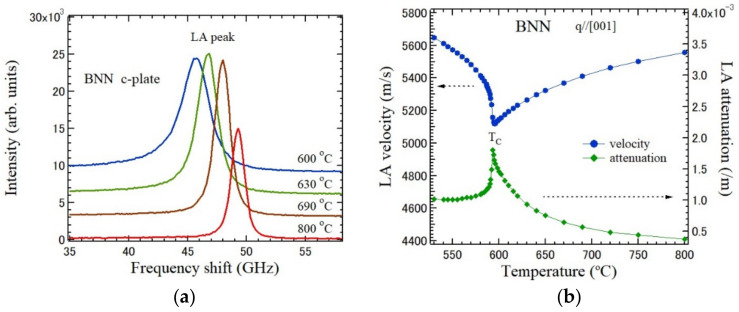
Temperature dependences of (**a**) Brillouin scattering spectra of a *c*-plate of BNN in a paraelectric phase, and (**b**) sound velocity and attenuation of the LA mode along the *c*-axis of BNN [61].

**Figure 8 materials-15-03518-f008:**
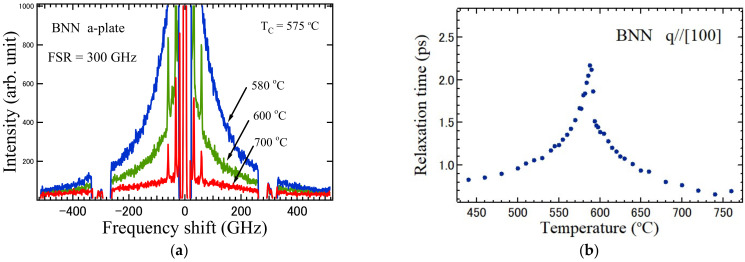
(**a**) Broadband Brillouin scattering spectra observed at backward scattering in a paraelectric phase of BNN [61]. A broad central peak appears in the vicinity of the Curie temperature. (**b**) Temperature dependence of relaxation time determined by the width of a broad central peak.

**Figure 9 materials-15-03518-f009:**
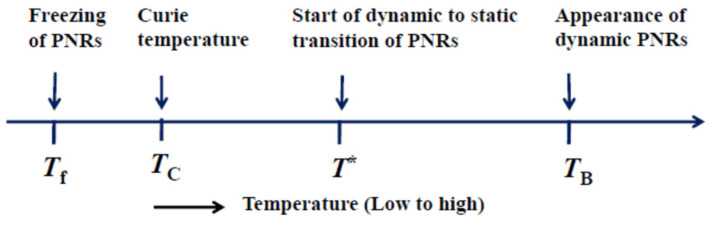
Characteristic temperatures of relaxor ferroelectrics. Polar nanoregions (PNRs) play a main role.

**Figure 10 materials-15-03518-f010:**
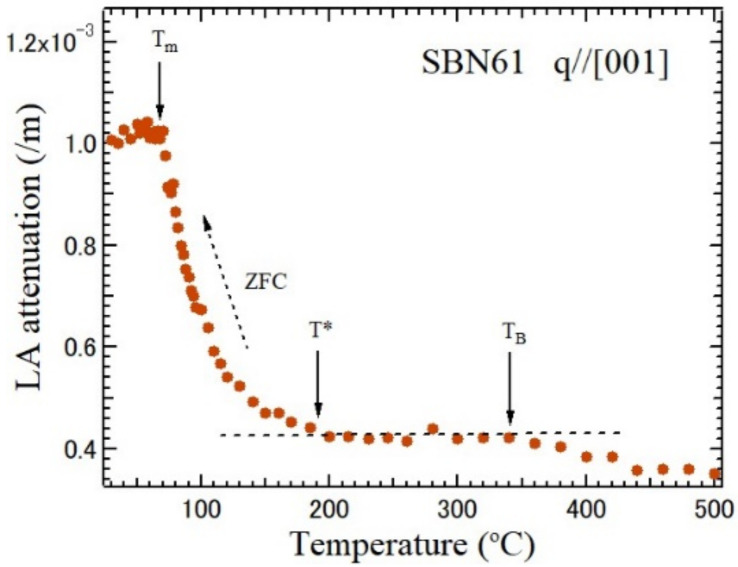
Temperature dependence of LA attenuation of SBN61 observed on zero-field cooling (ZFC), where *T*_B_ = 350 °C, *T** = 190 °C, and *T*_m_ = 72 °C [70].

**Figure 11 materials-15-03518-f011:**
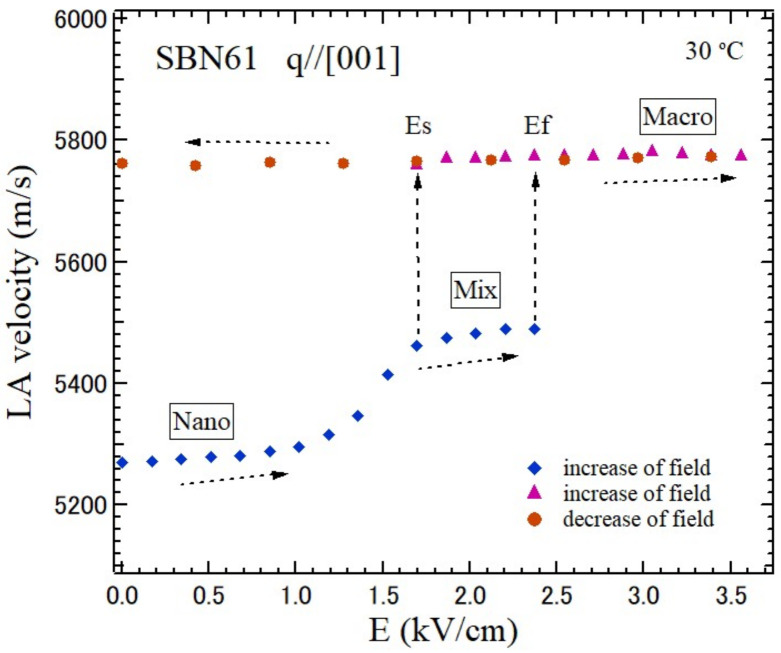
Variation of the LA velocity of SBN61 by the application of the electric field along the *c*-axis at 30 °C. By the application of the electric field along the *c*-axis to a (Nano) nanodomain state, (Mix) mixed, and (Macro) macrodomain states appeared [70].

**Figure 12 materials-15-03518-f012:**
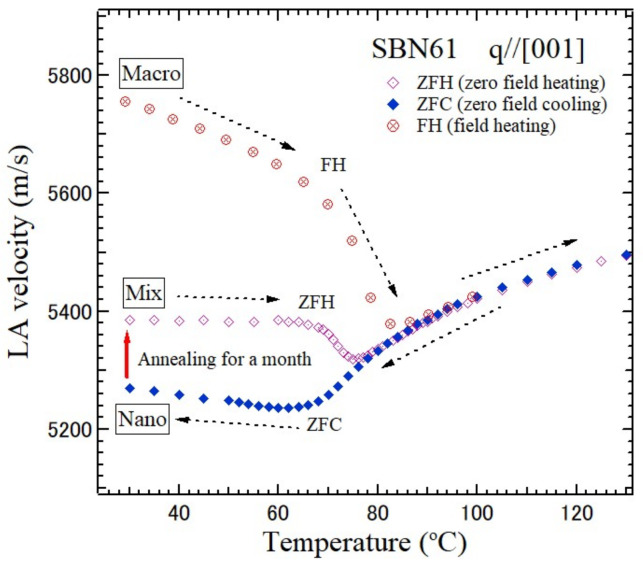
Variation of the LA velocity of SBN61 in the “Nano” state during ZFC without poling, the “Mix” state during zero field heating (ZFH) after a month at room temperature, and the “Macro” state during FH [70].

**Figure 13 materials-15-03518-f013:**
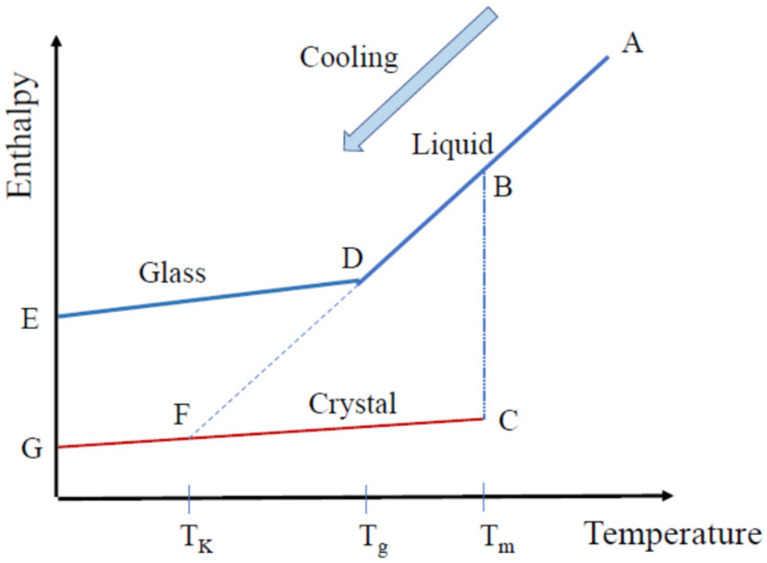
Temperature dependence of the enthalpy for the liquid (AB), supercooled liquid (BD), glass (DE), and crystalline (CG) states, where *T*_m_, *T*_g_, and *T*_K_ are the melting, glass transition, and Kauzmann temperatures, respectively.

**Figure 14 materials-15-03518-f014:**
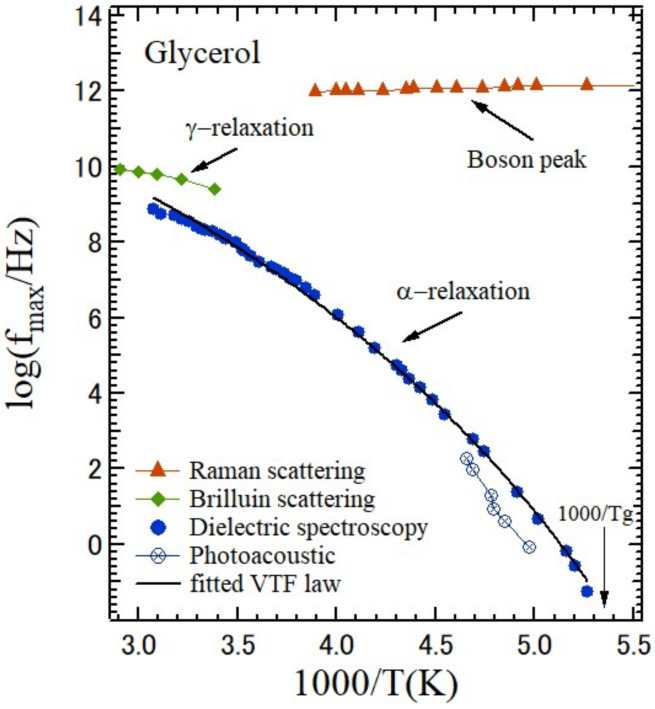
Relaxation processes and boson peaks of liquid and supercooled liquid states of glycerol. The main structural relaxation, α-relaxation, obeys the Vogel–Tammann–Fulcher (VTF) law [77,78,79,80].

**Figure 15 materials-15-03518-f015:**
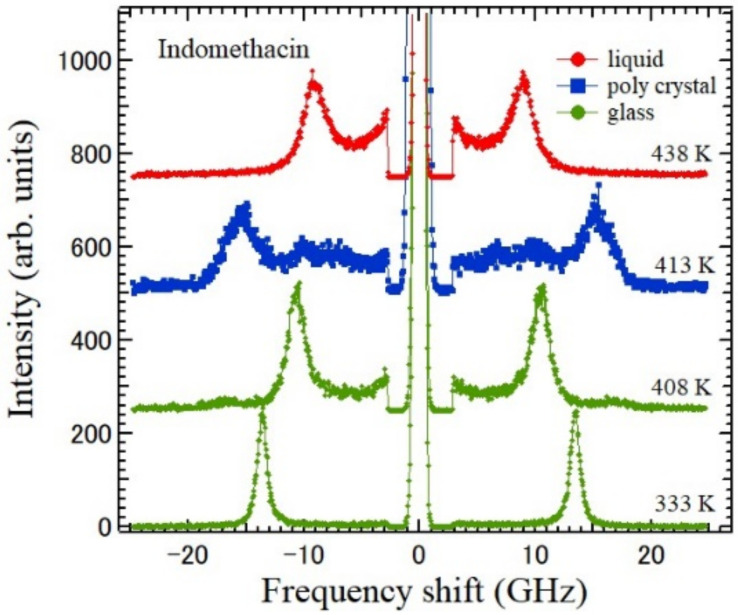
Temperature dependence of Brillouin scattering spectra of the liquid, glass, and polycrystalline states of indomethacin. The glassy states at 333 and 408 K, polycrystalline phase at 413 K after crystallization from glass, and liquid states at 438 K [95].

**Figure 16 materials-15-03518-f016:**
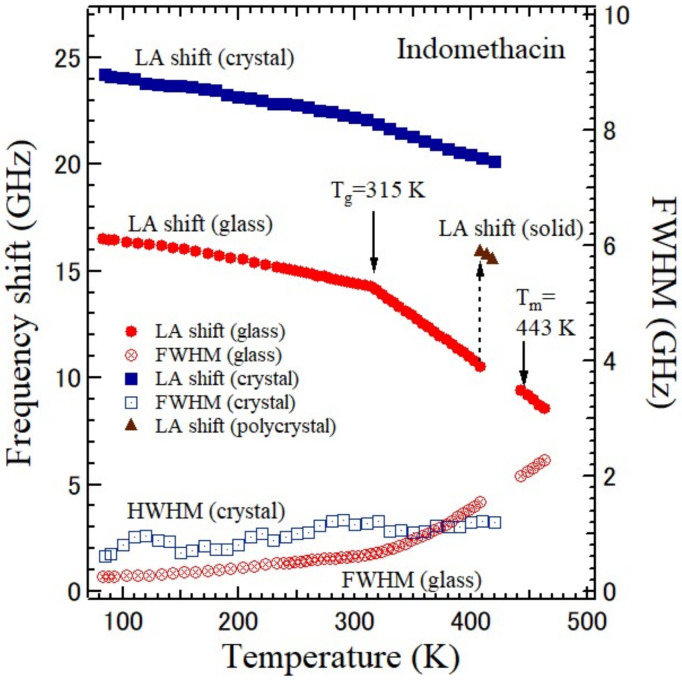
Temperature dependences of the LA frequency shift and FWHM of a γ-form indomethacin crystal with a triclinic structure, glass, and liquid states. The polycrystalline state created by the crystallization from a glass state at 410 K is also plotted [95].

**Figure 17 materials-15-03518-f017:**
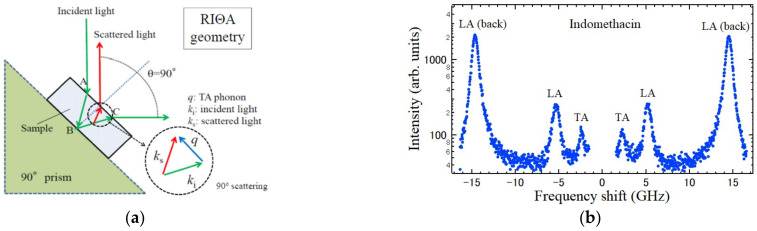
(**a**) Scattering geometry of the reflection-induced ΘA scattering to observe TA modes at the backward scattering geometry. By the reflection of an incident light at the point B, right angle scattering is observed. (**b**) The Brillouin scattering spectrum of the reflection induced ΘA scattering of a glassy IMC. The LA peaks of backward scattering were also observed [95].

**Figure 18 materials-15-03518-f018:**
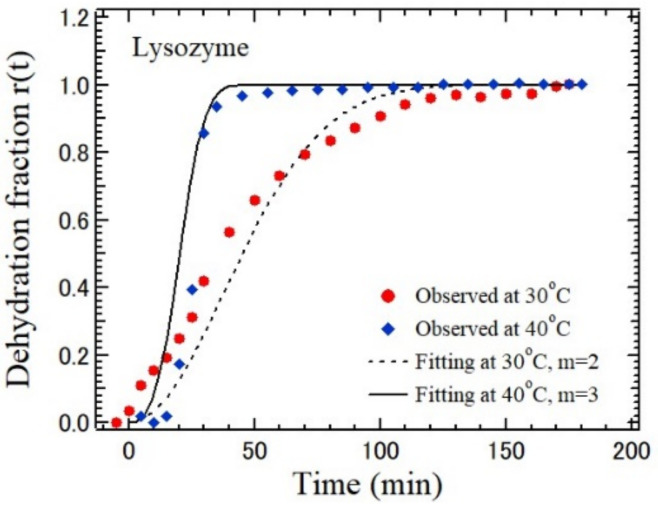
Time dependence of the normalized fraction of a tetragonal HEWL crystals at 30 and 40 °C [98].

**Table 1 materials-15-03518-t001:** History timeline related to Brillouin scattering in the early days.

Year	Event	Ref.
1899	A. Perot and C. Fabry developed the Fabry-Perot interferometer.	[8,9]
1912	P. Debye discussed acoustic oscillations in a box.	[10]
1922	L. Brillouin predicted inelastic light scattering by thermally excited acoustic waves.	[2]
1923	H. Nagaoka and T. Mishima developed the method to measure high resolution spectra using an echelon grating.	[11]
1926	L. Mandelstam independently predicted light scattering from thermally excited acoustic waves.	[12]
1930	E. Gross observed the inelastic light scattering from acoustic oscillations of liquids.	[7]

**Table 2 materials-15-03518-t002:** Elastic constants, *c*_ij_, of the γ-form IMC crystal at 296 K. Those of similar drug materials, aspirin, ibuprofen, are also shown for comparison [95].

**Elastic Constant (GPa)**	$C_{11}\;$	$C_{22}\;$	$C_{33}\;$	$C_{44}\;$	$C_{55}\;$	$C_{66}\;$
Indomethacin			17.89	5.44	3.39	
Aspirin [82]	11.29	12.10	10.67	3.67	2.68	4.41
Ibuprofen [83]	6.80				1.71	3.25

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
