# Peer review of "100th Anniversary of Brillouin Scattering: Impact on Materials Science"

_materials, 2022, doi:10.3390/ma15103518_

Round 1

Reviewer 1 Report

This review article introduced a brief history and summarized the techniques of  the Brillouin Scattering, followed by its application for various classes of materials.

I recommend publication of this review article after the following issues are resolved:

  1. The subscripts of many variables are not typed as subscripts.
  2. It seems that some figures were drawn by the author while the others appeared to have been copied from other sources where citation seems missing.
  3. Is there a way to outline a table  as timeline listing key results and the years and references? That will definitely help the readers. 

Author Response

Thank you very much for the review on my manuscript. I revise the subscripts and citation of figures.  Copyrights permissions have been obtained for figures. The manuscript has been revised by a native American scientist. The timeline of history on Brillouin scattering is added in Table 1. The revision is shown by red color in the revised manuscript.

Reviewer 2 Report

The manuscript presents an overview of the Brillouin scattering as an opportunity to study the structure and phase transitions of different materials. The development of spectroscopic methods and techniques base on Brillouin Scattering is presented at the beginning of the review. The main attention in the text is paid to studies of ferroelectrics and phase transitions in them. At the end of the review is presented the possibility of analysis of drug materials, as well as polymorphism, dehydration, and denaturation of proteins. The review will be useful both for researchers working on various materials not only listed above, but also for researchers using other methods such as Raman and THz spectroscopy.

I believe that the manuscript meets all the necessary requirements to be published as a review article. I have no further comments or remarks on the manuscript that need to be corrected.

In conclusion, I believe that the topic of this review article is in line with the topic of the special issue of the Materials MDPI journal and I recommend that it be published.

Author Response

(The authors gave the same response as above.)

Reviewer 3 Report

This is a review paper on Brillouin scattering which is common technique to examin various materials and elementary excitations  using quasi-elastic scattering
The authors gives a  the history, some technical background  and Brillouin scattering  and goes over some of the results and specific
materials that have been studied such ferroelectric materials, glasses, and proteins. Overall it is a nice brief review I suggest for publication. The author should go over the paper again for some wording issues but this is minor.

Author Response

(The authors gave the same response as above.)
